# SARS-CoV-2 in Mexico: Beyond Detection Methods, Scope and Limitations

**DOI:** 10.3390/diagnostics11010124

**Published:** 2021-01-14

**Authors:** Cynthia Martinez-Liu, Natalia Martínez-Acuña, Daniel Arellanos-Soto, Kame Galan-Huerta, Sonia Lozano-Sepulveda, María del Carmen Martínez-Guzmán, Ana Maria Rivas-Estilla

**Affiliations:** 1Department of Biochemistry and Molecular Medicine and Hospital Universitario “Dr. Jose E. Gonzalez”, Universidad Autonoma de Nuevo León, Monterrey 64460, Mexico; cmartinezli@uanl.edu.mx (C.M.-L.); nmartinez.me0120@uanl.edu.mx (N.M.-A.); daniel.arellanosst@uanl.edu.mx (D.A.-S.); kame.galanhr@uanl.edu.mx (K.G.-H.); slozano.me5017@uanl.edu.mx (S.L.-S.); 2Instituto Tecnologico de Tlahuac II, Tecnologico de Mexico, Mexico City 13550, Mexico; martinez_carmen@ittlahuac2.edu.mx

**Keywords:** SARS-CoV-2, IgG, antibodies, IgM, COVID-19, diagnostic test, immunoassay, ELISA, CLIA, rapid test, lateral flow assay, viral detection, immune host response

## Abstract

The new coronavirus that was first identified in December 2019 in Wuhan China, now called SARS-CoV-2, which causes the disease called COVID-19, has spread from China to the entire world in a few months. Due to its contagious potential (R0: 5.7) and because there is still no effective treatment to stop the infection, and a vaccine for prevention it is not yet available to the general population, COVID-19 is currently considered a global health problem. The need to implement sensitive methods for the identification of individuals with COVID-19 has led to the development of different molecular and immunological tests. The importance of a timely and accurate diagnosis is essential to determine the course of the pandemic. The interpretation of the results obtained by each test as well as the factors that affect these results have not been fully described. In this review, we describe and analyze the different SARS-CoV-2 detection methods that have been performed in Mexico and are available worldwide, outlining their strengths and weaknesses. Further, a broader perspective of the correct use and interpretation of the results obtained with these diagnostic tools is proposed to improve the containment strategy and identify the true impact of the pandemic.

## 1. Introduction

Coronavirus disease (COVID-19) is a respiratory disease caused by severe acute respiratory syndrome coronavirus 2 (SARS-CoV-2). Since its appearance in Wuhan China in December 2019, SARS-CoV-2 has caused to date (13 November 2020) 53,211,792 infections and 1,300,076 deaths worldwide, while Mexico reported 97,056 deaths and 991,835 infections on the same date, making it the country with the fourth highest number of deaths from this virus (reported by the database of the Center for System Science and Engineering at Johns Hopkins University) presenting a mortality rate of 9.8% (13 November 2020). Due to the great global impact caused by COVID-19, the World Health Organization declared a global health emergency on 30 January 2020.

SARS-CoV-2 is an enveloped virus with a positive polarity RNA genome and is approximately 90 nm^3^ in diameter size; it belongs to the *Coronaviridae* family of the Betacoronavirus genus. The genome of SARS-CoV-2 is approximately 29.7 kb long, which encodes a spike (S), an envelope (E), a membrane (M), nucleocapsid (N) proteins, and six accessory proteins (3, 6, 7a, 7b, 8, and 9b), and comprises a large open reading frame (ORF) encoding polyproteins pp1a and pp1ab, which are further cleaved into 16 nonstructural proteins [1].

During the infection process, SARS-CoV-2 interacts with the host cell through a receptor binding domain belonging to the S1 subunit protein S called RBD. SARS-CoV-2 RBD binds to the angiotensin-receptor converting enzyme 2 (ACE2) which is present in the epithelial cells of the respiratory tract and in many other extrapulmonary tissues including heart, kidney, endothelium, and intestine [2]. After binding, the virus is internalized into cells, initiating its replicative cycle [3].

Due to the wide distribution of the ACE2 receptor in the human body, multiple pathologies have been observed in addition to the lung damage related to SARS-CoV-2 infection, such as myocardial injury and arrhythmias [4], pancreatic injury [5], brain injuries, and dysregulated neurochemical activity [6], causing greater than expected complications and long-term sequelae in patients and even increasing the mortality rate in severe cases. 

Due to the severity of symptoms that COVID-19 can cause and the possible clinical consequences that are just being determined in patients in recovery, in addition to the great impact COVID-19 has already had on the population and economy in the last year, the correct identification of people carrying SARS-CoV-2 is required as one of the key points for the containment of this pandemic.

In order to fulfill this need, multiple methods for identifying SARS-CoV-2 infection have been developed. The accelerated progress and commercialization of this range of tests has generated confusion and uncertainty in their use and interpretation of their results. In this context, this article aimed to point out the differences in the tests and outline the conditions in which each of the tests that are available in Mexico can generate a reliable result. In addition, it addresses the guidelines for the use of each test and the new technologies that are being used in Mexico to speed up the identification of COVID-19 cases to improve the containment strategies of the pandemic and implement a safe economic reactivation in Mexico and countries with similar socioeconomic scenarios.

## 2. Symptoms and Transmission Routes of COVID-19

COVID-19 presents as an atypical pneumonia causing different symptoms including fever, fatigue, dry cough, myalgia and dyspnea and causing complications and the highest mortality rate in patients with different comorbidities, such as diabetes, hypertension, obesity, and other related immune system diseases [7]. However, most individuals develop the disease mildly or even asymptomatically.

COVID-19 symptoms can be confused with the clinical manifestations of other respiratory diseases (influenza, respiratory syncytial syndrome, etc.) or febrile infectious diseases (Dengue, Chikungunya, and Zika), so its diagnosis relies on the availability of systems that allow specific detection to determine if a person suffers from an infection by this virus.

SARS-CoV-2 can be transmitted through close contact with contaminated secretions from people infected with the virus that are expelled when they cough, sneeze, or talk (direct transmission) or by contact with contaminated surfaces (indirect transmission). For SARS-CoV-2, the R0 (the value that refers to the contagious capacity of the pathogen) has been calculated to be 5.7, indicating a rapid spread in the population, much greater than what had been estimated at the beginning of the pandemic [8].

Therefore, accurate and timely diagnosis of infected people and confinement is required for proper management, containment of outbreaks, screening of the population, and determination of public health strategies. The current criteria used for the identification of positive cases have the main limitation that mild infections and asymptomatic cases are undiagnosed, causing failure of the prevention measures due to ignorance of an active infection by asymptomatic carriers, triggering the excessive spread of the virus in the population.

The design of integrative strategies for case identification, such as the proper use and interpretation of diagnostic tests, is one of the most urgent needs. 

## 3. Diagnostic Tests for SARS-CoV-2

Tests to diagnose COVID-19 are based on direct or indirect detection of the virus through different techniques. The direct detection tests identify the virus by its genome, using molecular methods such PCR-based assays or by its viral proteins using rapid antigen detection tests through anti-SARS-CoV-2 antibodies. Detection tests are indirect when elements of the response to viral infection from the host, such as specific anti-pathogen antibodies, are identified by immunoassays such as enzyme-linked immunosorbent assay (ELISA), chemiluminescent immunoassay (CLIA), lateral flow assay (LFA), etc. In both types of tests, molecular principles (targeting of nucleic acids) or immunological principles (using the antigen–antibody binding) are used to carry out the detection as shown in Figure 1.

There are two types of medical diagnostic tests depending on the place where they take place, laboratory tests and point of care (POC) tests. The first ones are sensitive, have high throughput, and are performed in a controlled environment with specialized equipment and personnel; the POC tests are rapid tests that use the same biological principles as those used in the laboratory tests but with minimal manipulation of the sample in order to obtain results in less time (approximately 15 min) and without requiring specialized equipment.

Both types of diagnostic tests are evaluated using metrics to determine their detection limits, their reliability, and the error rates they present [9]. To assess the reliability of a clinical diagnostic test, different parameters have been used, such as sensitivity, specificity, positive predictive value (PPV), negative predictive value (NPV) predictive value, and the likelihood ratio. Sensitivity and specificity are values that measure the ability of a test to correctly classify an individual as either diseased or disease-free [10]. These values are determined according to established formulas [11] that are calculated by comparing the results obtained by the test to the diagnostic test considered as the reference standard.

Based on this, laboratory tests (because they are performed in controlled environments and have the calibration standards available for the specialized equipment) generally have higher sensitivity and specificity and are established as the gold standard tests for the clinical diagnosis of different pathologies; however, they are not always the most practical or the easiest to obtain for the population. Rapid tests are normally used to support a clinical diagnosis in hard-to-reach places that do not have the necessary facilities to complete the necessary laboratory tests.

## 4. What Are Molecular Tests and How Do They Work to Identify COVID-19?

The RT-qPCR assay has been used throughout the world as the gold standard diagnostic test for the confirmation of SARS-CoV-2. Currently, the only approved protocols for the diagnosis of COVID-19 in Mexico are based on the direct detection of the viral genome by this molecular assay. 

The RT-qPCR assay consists of the detection and amplification of one or more of the specific genes of the virus, including RdRp (RNA-dependent RNA polymerase), N (nucleocapsid protein), E (envelope protein), S (spike protein), ORF1ab nsp10 (non-structural protein 10), ORF1b nsp14 (non-structural protein 14), and human ribonuclease P (RNase P) gene. RNase P gene serves as an internal amplification control and indicator of specimen adequacy. Detection of each target may require a separate polymerase chain reaction (PCR) and the number of PCRs required depends on the number of objectives that need to be amplified for each specimen tested; however, in order to simplify the detection, multiplex tests have been developed to detect different genes in a single reaction.

Different protocols have emerged around the world; however, only seven have been recommended by the World Health Organization (WHO) as diagnostic protocols (see Table 1). The most common specimens used to perform this confirmatory test are nasopharyngeal obtained by swabbing; however, oropharyngeal samples, sputum, and bronchoalveolar washings are also used [12,13]. As an alternative, the use of saliva as a less invasive specimen for the detection of SARS-CoV-2 by RT-qPCR has been studied [14,15]. Despite the fact that many studies agree with the similarity of viral detection by PCR in oropharyngeal and nasopharyngeal swabs with saliva [16], in Mexico, its use as a specimen for the diagnosis of COVID-19 by RT-qPCR is not authorized.

The use of various specimens for the detection of COVID-19 such as tears and conjunctival secretions [24,25], urine, and feces has been explored; however, they have not yet been established as samples for use in viral detection by RT-qPCR [26].

For the handling of potentially infectious samples from patients who may have COVID-19, protocols of management and testing were determined by the WHO, such as for the correct collection and transport of specimens, use personal protective equipment, and security procedures that each laboratory and interested party involved in COVID-19 tests must take into account. They also defined protocols for the subsequent viral RNA extraction and molecular testing. In order to facilitate the diagnosis of COVID-19, different companies have developed and commercialized kits, which contain all the reagents necessary to perform the RT-qPCR tests.

The assertiveness of the diagnosis of the commercially available kits has been discussed in different articles, and although the detection limits do not correspond to those reported by the supplier, it has been possible to corroborate the tests’ usefulness and to use them as diagnostic methods for detection [27,28].

Due to differences in the detection limits of molecular tests (which restricts this type of detection) and the interference of other factors that affect the performance of these tests in obtaining a correct result, accessory tests have been developed for the indirect identification of SARS-CoV-2 infection.

## 5. What Are Immunological Assays and How Do They Work to Detect COVID-19?

Immunological assays are indirect viral detection methods based on the principle of antigen–antibody binding. There are two detection strategies for this type of assay: host antibody detection using their labeled antigen and viral antigen detection using a labeled specific antibody.

Serological tests are used as part of the monitoring schemes for the pandemic and the clinical evolution of the patient. These are a type of immunoassay focused on host antibody detection. These tests are oriented to determine the seroprevalence of the population against SARS-CoV-2. The antibodies profiles of patients with COVID-19 has been described. IgA appears on average 13 days’ post symptom establishment [29] and IgM appears average 10 days’ post symptom establishment and persists until day 42 [30]. IgG seroconversion has been reported on average on the 13th day after the onset of the first symptoms [31] and was detected up to 5 months after the notification of the positive result by RT-qPCR for SARS-CoV-2 [32].

It should be mentioned that the number of days of seroconversion varied in the different studies due to the differences in the sensitivity of the methods used to measure seroconversion. However, most of the patients with COVID-19 show seroconversion by 3 weeks’ post symptom establishment. In addition, IgG presence has been related to neutralizing activity; that is why detection of this antibody class is a prerequisite for the plasmapheresis protocol used as therapy to treat patients with COVID-19, and the reason why its detection has become important. There are different formats of serological tests, the most popular are ELISA, CLIA, and rapid LFA antibody-based tests.

ELISA is a popular analytical biochemistry assay that uses a solid-phase enzyme immunoassay (EIA) to detect and quantify the presence of analytes, including antibodies, antigens, proteins, etc., in liquid or wet samples. ELISA uses a different capture system, where the analyte of interest is immobilized and subsequently labeled with a molecule that produces a detectable signal. ELISA is used to detect viral elements or specific antibodies against SARS-CoV-2 in serum and plasma. Different companies have commercialized this type of assay; however, the limited access and availability in some countries to ELISA, such as in Mexico, has reduced its application in the routine protocols implemented for the management and monitoring of COVID-19 (18 May 2020 was when the first serological test was approved in Mexico for distribution and commercialization).

CLIA is an immunoassay technique (similar to ELISA) where the label of the analytical reaction is a luminescent molecule, created either directly by using luminophores markers or indirectly using enzyme markers. CLIA is more sensitive than is ELISA and currently is used for screening of antibodies against SARS-CoV-2. Several companies have developed and commercialized ELISA and CLIA kits around the world to measure, under laboratory conditions, the presence of antibodies, mainly IgG and IgM, using different antigens (usually the viral S and N structural proteins) as detecting molecules.

Rapid lateral flow antibody-based tests or LFAs are paper-based chromatographic tests that already contain the pre-loaded reagents to generate the detection reaction. LFAs use the same principles as other serological tests; however, their format is smaller and more portable and by design they do not need to be performed in a laboratory environment.

LFAs consist of different components and as the sample flows through the strip, antigen–antibody interactions occur. The strip also contains a marker (usually gold nanoparticles) that emits a signal when the interaction has occurred. Then, the results are viewed in the detection window, showing if antibodies are present in the sample, generating a result in approximately 15 min.

Different companies developed and commercialized rapid test formats for the identification of IgG and IgM antibodies to identify antibodies against COVID-19. Research groups around the world have been given the task of evaluating the performance of these tests [33] and comparing the three serological methods already mentioned. They have observed that LFAs have a lower sensitivity than ELISAS and CLIAS [34,35]. For this reason, LFAs are suggested for monitoring the immune response in the population, but they are not indicated as diagnostic tests to identify current SARS-CoV-2 infections.

To validate these tests and use them as possible complementary diagnostic methods for the indirect detection of SARS-CoV-2, different studies have been carried out where the sensitivity and specificity of each test was determined. Different authors have combined all this information in review papers, where the characteristics of each serological test are fully explained [36,37,38].

However, it remains to be verified with the use of various cohorts and experiments if these types of tests that measure immune response could be used as detection methods for COVID-19 or can only continue to be implemented as immunological monitoring tests.

## 6. Interpretation of Results Obtained by SARS-CoV-2 Detection Methods in Mexico and Their Pitfalls

Confirmatory tests by RT-qPCR are the official tests validated by the WHO and different organizations to diagnose COVID-19; however, the presence of false negatives in these tests has been reported. The accuracy of RT-qPCR results is related, like in other diagnostic tests, to the amount of viral genetic material available to be detected (limit of detection; LoD). Factors such as the time at which the biological sample is taken (i.e., at a very early or late stage of the infection when the viral load is low) and the quality (degradation of genetic material) and amount of the sample influence the accurate result.

Another explanation for the false negatives is the mutations generated by the virus in its genome (as has been reported in other viruses) [39,40], changing the recognition sequence of the primers so that amplification may not occur and false negatives are obtained. 

It has also been suggested that the symptoms are important to determine the time at which to perform the molecular tests to obtain accurate results. A relationship has been observed between the patient's symptoms and the time in which the detection of the viral genome can be carried out, where in individuals with symptoms, viral detection can be carried out for a longer period than in asymptomatic patients [41].

Immunological tests are currently restricted for use as confirmatory diagnostic tests for COVID-19 and are only used as monitoring tests of the immune response that indicates the presence of antibodies against COVID-19. This restriction is given by the principle of the test, where the results are subject to the immune response of each individual. Several studies have reported that the antibodies generated against SARS-CoV-2 take approximately 10 to 14 days after the first symptoms appear and are detectable until the 25th day, therefore, the time window when the serological test is performed and the sensitivity of the test determines the accurate result [42,43]. Efforts have focused on determining the pattern of immunity generated by this disease; however, this depends on each population and subject.

On the other hand, it has been observed that not all individuals with COVID-19 generate antibodies against the virus, but rather develop only memory T-cells that act to counteract the infection [44,45]. The detection of antibodies by serological tests in this population with only specific T-cells against SARS-CoV-2 generates a false negative in a serological test.

Differences have been reported in the kinetics of antibodies against SARS-CoV-2 in individuals with the presence and absence of symptoms, suggesting that asymptomatic patients develop a weaker immune response than do symptomatic patients [46]; however, no differences have been found between time and the seroconversion of IgA, IgM, and IgG [47,48], therefore, serological tests may be performed in the same time window and accurate results are obtained for both.

In conclusion, the application and the certainty of the results of the molecular and serological tests depend on the pattern of viral behavior and the seroconversion and prevalence of the host’s antibodies during the infection (see Figure 2).

In order to determine the veracity of the results, different agencies around the world have been validating the protocols and commercial tests for use as diagnostic or monitoring tests. 

In the United States and its territories, the Food and Drug Administration (FDA) has validated and approved 149 molecular diagnostic tests, 35 molecular tests for the detection of viral nucleic acids, 4 immunological tests for the detection of viral antigens, and 40 serological tests for the detection of antibodies against SARS-CoV-2 (28 August 2020).

In Mexico, the agency in charge of validating and regulating diagnostic tests for COVID-19 is InDRe (National Institute for Epidemiological Diagnosis and Reference “Dr. Manuel Martínez Báez”). InDRe developed and transferred a molecular detection test to identify SARS-CoV-2 to public state diagnostic laboratories. For the commercialization and use of molecular diagnostic tests in the private sector, InDRe has evaluated different commercial kits, of which 50 are approved in Mexico to be used as confirmatory diagnostic methods for COVID-19 (as of 13 November 2020).

Serological tests in Mexico are not approved as diagnostic tests for COVID-19; however, InDRe has validated 39 serological tests (13 November 2020) in different formats for the detection of IgG and IgM antibodies against this virus (see Table 2), which are used exclusively to monitor neutralizing antibodies in plasma donors for the treatment of critically ill patients with this disease.

InDRe also recently approved (13 November 2020) two rapid tests for the detection of SARS-CoV-2 by viral antigens; however, these tests have not been authorized as diagnostic tests in Mexico and their use does not replace the diagnostic test for genome detection by RT-qPCR.

To facilitate decision-making in the context of detection and diagnosis of COVID-19, this article lists each of the components involved in the infection process and the viral characteristics, placing into perspective how to use each diagnostic tool (see Figure 3).

## 7. Future Prospect: Use of Point-Of-Care Tests as Tools to Detect COVID-19

Using tests at the point of care, or POC tests, have become popular worldwide to have accurate results in less time and without using specialized equipment and laboratory personnel. POC tests are located at the site of first contact of the patient in the health center and provide in a short period of time (approximately 15 min) relevant information of their condition, allowing for a quick triage.

These POC tests have a variety of characteristics: they are portable, easy to use, employ a minimum sample volume, and they can compete in sensitivity and specificity with conventional laboratory tests. POC tests use different methods for the detection of analytes in complex heterogeneous biological samples such as blood, saliva, urine, etc., and they are available in multiple easy-to-use formats. Some POC tests require the use of portable reading devices (semi-quantitative-quantitative), allowing for the collection of data, the monitoring of results, and the traceability of diseases. 

Distinct POC test formats have been developed using biosensors, such as chips and paper-based chromatographic formats for the detection of pathogens that cause infectious diseases (human immunodeficiency virus (HIV), influenza, *Treponema pallidum* (syphilis), Hepatitis C, etc.), and multiple devices are currently commercially available, so they could be a practical solution for the early detection and improvement in monitoring of COVID-19.

POC tests for the identification of SARS-CoV-2 by viral antigens in LFA formats have been developed and approved by the FDA. These tests are based on the qualitative detection of SARS-CoV-2 by identifying the proteins in its envelope, which are recognized by immobilized antibodies in the lateral flow device. The result is interpreted without specialized equipment and is available in 30 minutes. Due to the reported sensitivity and specificity (91.4% and 99.8% Panbio COVID-19 Ag Rapid Test; 96.7% and 100% Sofia 2 SARS antigen fia test, respectively) and their easy use, they are positioned as the future of diagnostic tests for COVID-19. In Mexico, two of these tests, mentioned previously in the paper, have been validated for use; however, they are not yet considered diagnostic tests per se, and they are only authorized as follow-up and diagnostic support tests since the sensitivity and specificity values have not been evaluated in the Mexican population.

The development of POC tests using RT-LAMP technology for the diagnosis of COVID-19 have also been reported. Loop-mediated isothermal amplification (LAMP) is a fast and specific technology for DNA amplification that uses four to six primers that bind to six regions of the DNA fragment to be amplified using a single temperature (generally 65 °C), generating a structure stem-loop during hybridization [53]. POC tests in different formats have been developed using LAMP technology coupling one or more steps for reverse transcription (RT-LAMP) for the identification of different viruses (with RNA genomes) in the clinical area [54,55,56].

This technology was compared with the gold standard and a high specificity was observed that can compete with that of laboratory tests RT-qPCR (99–100%) but its sensitivity still needs to be improved as it is approximately 86% [57,58]. Currently, the FDA approved these POCs as an alternative for the identification of SARS-CoV-2 in pharyngeal samples; however, this technology is not available in Mexico. The establishment of the use of these devices in the country requires a validation process by inDRE and their application in the public health system is not feasible due to the cost of their implementation.

Compared to laboratory tests, POC tests have some disadvantages that could compromise the veracity of the results, such as the lack of calibration of the equipment used, the use of adequate quality controls, the incorrect handling of reagents or equipment by the personal, different conditions that can affect the performance of each technology, etc. [59]. However, the use of reagents that are stable under various conditions and the generation of manuals and reagents for their calibration are improvements to be made in POC tests to avoid these nonconformities.

The use of POC test formats for the detection of COVID-19 would allow for the rapid identification and containment of positive cases, generating a decrease in cases, an early follow-up of patients, a decrease in saturation in hospitals, an acceleration of triage, and a tool for monitoring in different sectors that allows for safe economic reactivation while minimizing the negative impact of the pandemic. POC tests could be the answer to all the deficiencies presented in the diagnostic tests for the detection and classification of patients with COVID-19.

## Figures and Tables

**Figure 1 diagnostics-11-00124-f001:**
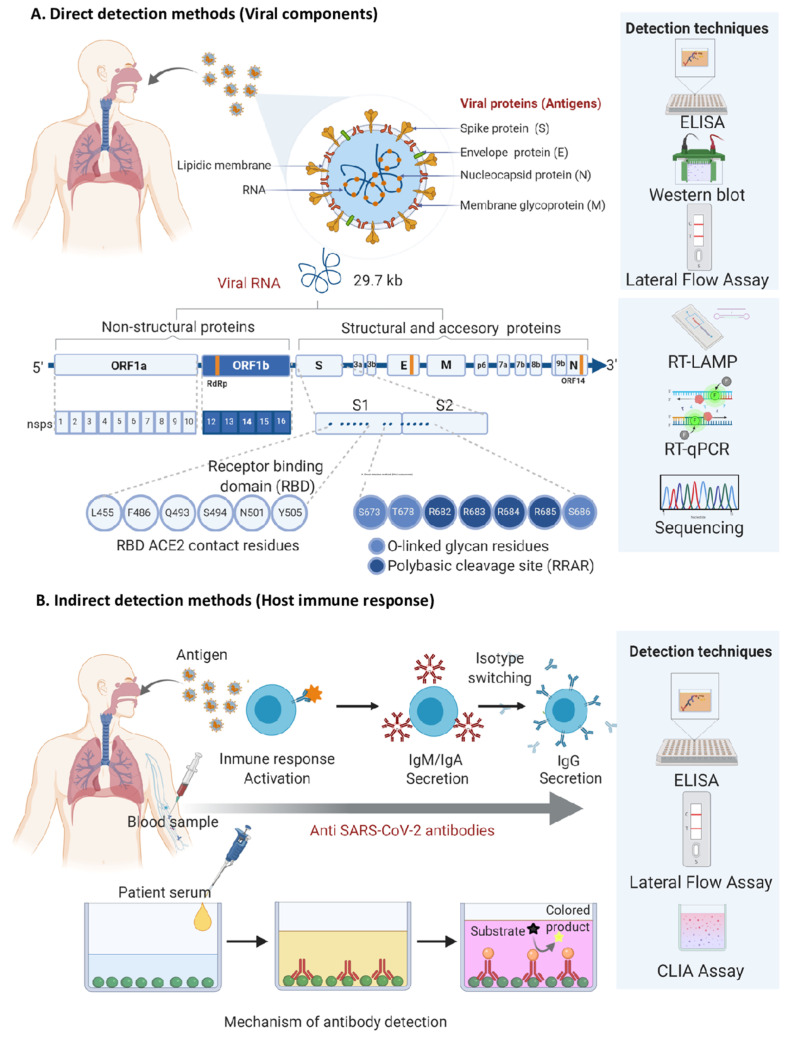
Fundamental elements for the recognition of severe acute respiratory syndrome coronavirus 2 (SARS-CoV-2) used in the diagnostic tests validated in Mexico. (**A**) Direct detection methods target viral components such as viral proteins and genome. RT-qPCR and new generation sequencing are the main techniques used for viral genome detection; while ELISA, LFA, and Western Blot are used for viral protein detection. A positive result indicates that the virus is present in the analyzed sample and may be used to detect currently infected patients. The RT-LAMP technology also focuses on the detection of the viral genome, but its use in Mexico has not yet been validated. (**B**) Indirect detection methods are directed against anti-SARS-CoV-2 antibodies produced by the host´s immune cells after an infection event. This strategy uses recombinant viral proteins as bait to capture the host´s antibodies. Due to the time needed for antibody production, a positive result means that the person was infected at some point in the past, but may not be currently infected. Antibodies can be detected by LFA, ELISA, and CLIA. CLIA: Chemiluminescent immunoassay; LFA: lateral flow assay; ELISA: enzyme-linked immunosorbent assay; RT-qPCR: quantitative reverse transcription polymerase chain reaction; RT-LAMP: reverse transcription loop-mediated isothermal amplification.

**Figure 2 diagnostics-11-00124-f002:**
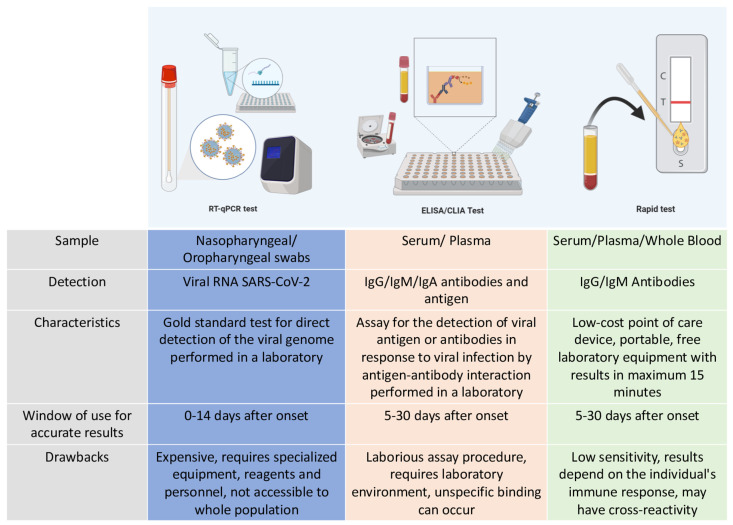
Differences between molecular and serological tests performed in Mexico. The characteristics of each detection method for SARS-CoV-2 infections and their different properties are compared, specifying the time window of their effectiveness to produce an accurate diagnosis. The assessment of all these characteristics allows for the correct choice of the detection method.

**Figure 3 diagnostics-11-00124-f003:**
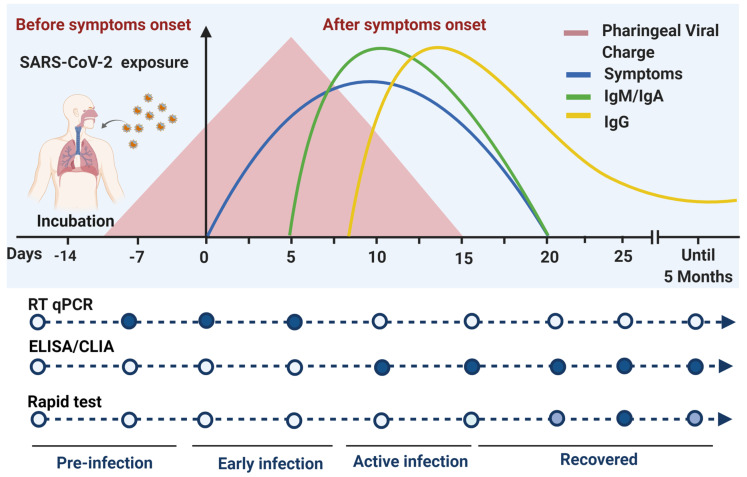
Timeline for the accurate detection of COVID-19 with authorized tests in Mexico. Due to its nature, detection of SARS-CoV-2 and diagnosis of COVID-19 implies the appropriate combination of detection methods and medical supervision. The most important point for making decisions in a probable case validation is the symptom establishment day (day 0). Depending on the time between the symptom onset and when the patient arrives at the hospital, applicability of the detection methods can change. Viral RNA can be detected by RT-qPCR on pharyngeal samples in the first days after symptoms establishment. Antibodies can be produced as soon as 5 days after symptoms establishment but most of the infected patients take until a month to produce detectable levels of antibodies. CLIA: chemiluminescent immunoassay; ELISA: enzyme-linked immunosorbent assay; RT-qPCR: quantitative reverse transcription polymerase chain reaction.

**Table 1 diagnostics-11-00124-t001:** Protocols validated by the WHO for detection of SARS-CoV-2 by real-time RT-PCR.

Institution	Gene Targets	LoD	Reference
China CDC, China	ORF1ab and N	100 RNA copies/µL	[17]
Institut Pasteur, Paris, France	RdRp IP2 and RdRp IP4	10 copies RNA genome	[18]
Charité (Germany)	RdRP, E and N	E: 3.9 copies RNA genome	[19]
		RdRp: 3.6 copies RNA genome	
US CDC, USA	N1 and N2	100 RNA copies/µL	[20]
HKU, Hong Kong SAR	ORF1b-nsp14 and N	Not specified	[21]
National Institute of Health, Thailand	N	Not specified	[22]
National Institute of Infectious Diseases, Japan	ORF1a and S	Not specified	[23]

LoD: Limit of detection; CDC: Center for Disease Control and Prevention; HKU: University of Hong Kong; Hong Kong SAR: special administrative region of Hong Kong.

**Table 2 diagnostics-11-00124-t002:** Serological tests validated by the National Institute for Epidemiological Diagnosis and Reference “Dr. Manuel Martínez Báez” (InDRe) in Mexico for the detection of antibodies against SARS-CoV-2.

Name	Manufacturer	Technology	Target	Sensitivity (%)	Specificity (%)
Architect SARS CoV-2 IgG	Abbott Laboratories Inc	CLIA	IgG	100 ^2^	99.6 ^2^
2019-nCoV Specific Test (IgG and IgM antibody determination kit)	Beijing Diagret Biotechnologies Co., Ltd	LFA	IgG/IgM	83 ^1^	93 ^1^
COVID 19 IgG-IgM Cassette	Hangzhou Biotest Biotech Co. Ltd.	LFA	IgG	90 ^2^	100 ^2^
			IgM	100 ^2^	98.8 ^2^
Certum 2019-nCov IgG/IgM Rapid Test	Hangzhou AllTest Biotech Co. Ltd.	LFA	IgG	99.9 [49]	98 [49]
			IgM	90.9 [49]	97 [49]
Standard Q COVID-19 IgM/IgG Combo Test	SD Biosensor, Inc.	LFA	IgM	53.3 ^2^	100 ^2^
			IgG	73.3 ^2^	98.8 ^2^
Panbio COVID-19 IgG/IgM Rapid Test Device	Abon Biopharm (Hangzhou) Co.,Ltd	LFA	IgG/IgM	96.2 ^1^	100 ^1^
Novel Coronavirus 2019 nCoV IgG/IgM Test Kit (colloidal gold)	Genrui Biotech Inc.	LFA	IgG/IgM	90.74 ^1^	94.64 ^1^
COVID-19 Antibody Test Kit (SARSCoV-2) ACC100-CRDT-COVID19-KIT	Accutest Research Laboratories México, S.A. de C.V.	LFA	IgG/IgM	89.67 ^1^	99.33 ^1^
EdinburghGenetics COVID-19 Colloidal Gold Immunoassay Testing Kit, IgG/IgM Combined	EdinburghGenetics Limited	LFA	IgG/IgM	98.43 ^1^	99.31 ^1^
Elecsys Anti-SARS-COV-2. Cobas^®^	Roche Diagnostics GmbH	CLIA	IgG	99.5 ^1^	99.44 ^1^
Anti-SARS-CoV-2 ELISA (IgA)	Euroimmun Medizinische Labordiagnostika AG	ELISA	IgA	93.3 [50]	80.0 [50]
COVID-19 IgG/IgM Test Cassette (Testsealabs^®^)	Hangzhou Testsea Biotechnology, Co., Ltd.	LFA	IgG	40 ^2^	93.8 ^2^
			IgM	73.3 ^2^	98.8 ^2^
WHPM COVID-19 /IgM /IgG Rapid Test	W.H.P.M., Inc.	LFA	IgG /IgM	76.7 ^2^	97.1 ^2^
2019-nCoV Ab IgM/IgG (Innovita)	Innovita (Tangshan) Biological Technology Co., Ltd	LFA	IgG/IgM	87.3 ^1^	100 ^1^
Innoscreen COVID 19 IgG/IgM Rapid test	Innovation Scientific Pty LTD	LFA	IgG/IgM	Not specified	Not specified
Liaison SARS-CoV-2 S1/S2 IgG	Diasorin, S.p.A	CLIA	IgG	64 ^1^	97.7 ^1^
MAGLUMI 2019-nCoV IgG (CLIA)	Shenzhen New Industries Biomedical Engineering Co., Ltd	CLIA	IgG	95.6 ^1^	96 ^1^
MAGLUMI 2019-nCoV IgM (CLIA)	Shenzhen New Industries Biomedical Engineering Co., Ltd	CLIA	IgM	89.89 ^1^	96.5 ^1^
COVID-19 lgG/lgM Rapid Test Cassette (Wb/S/P)	Hangzhou Clongene Biotech Co., Ltd.	LFA	IgG/IgM	Not specified	Not specified
Anti-SARS-CoV-2 ELISA (IgG)	Euroimmun Medizinische Labordiagnostika AG	ELISA	IgG	90 ^1^	100 ^1^
Anti-SARS-CoV-2 NCP ELISA (IgG)	Euroimmun Medizinische Labordiagnostika AG	ELISA	IgG	94.6 ^1^	99.8 ^1^
2019-nCoV IgG/IgM Rapid Test Device	Hangzhou Realy Tech Co., Ltd.	LFA	IgG/IgM	95.5 [51]	96.8 [51]
Vitros Anti-SARS-CoV-2 IgG	Ortho-Clinical Diagnostics, Inc.	CLIA	IgG	90 ^2^	100 ^2^
Vitros Anti-SARS-CoV-2Total Antibodies (IgA, IgM e IgG)	Ortho-Clinical Diagnostics, Inc.	CLIA	IgA/IgG/IgM	100 ^2^	100 ^2^
NADAL COVID 19 lgG/lgM TEST KIT (Test cassette)	NAL VON MINDEN GmbH	LFA	IgG/IgM	93.7 ^1^	99.1 ^1^
careUS COVID-19 lgM/lgG	Wells Bio, Inc.	LFA	lgM/lgG	93.6 ^1^	98 ^1^
WANTAI SARS-CoV-2 Ab Rapid Test	Beijing Wantai Biological Pharmacy Enterprise Co.	LFA	lgM/lgG	100 ^2^	98.8 ^2^
VIDAS SARS-CoV-2 IgG (9COG)	Biomerieux, S.A.	ELFA	IgG	96.6 ^1^	99.9 ^1^
VIDAS SARS-CoV-2 IgM (9COM)	Biomerieux, S.A.	ELFA	IgM	100 ^1^	99.4 ^1^
Cellex qSARS-CoV-2 IgG/IgM Cassette Rapid Test	Cellex Biotech (Suzhou) Co, Ltd.	LFA	IgG/IgM	20 [35]	100 [35]
SureScreen COVID-19 IgG/IgM Rapid test	Surescreen Diagnostics, Ltd	LFA	IgG/IgM	96.5 ^1^	99.67 ^1^
Diagnostics Kit (Colloidal Gold) for IgG/IgM Antibody SARS-CoV-2	Xiamen Wiz Biotech Co., Ltd.	LFA	IgG/IgM	Not specified	Not specified
Biocredit COVID-19IgG+IgM Duo Rapid Test	RapiGEN, Inc.	LFA	IgG	94.2 ^1^	98.7 ^1^
			IgM	100 ^1^	98.7 ^1^
QikTech COVID-19 IgG/IgM Antibody Test	Lusys Laboratories, Inc.	LFA	IgG/IgM	Not specified	Not specified
Acon^®^ SARS-CoV-2 IgG/IgM Rapid Test	Acon Biotech (Hangzhou) Co., Ltd.	LFA	IgG/IgM	99.1 ^1^	98.2 ^1^
NOVA test	AtlasLink (Beijing) Technology Co., Ltd.	LFA	IgG	90 ^2^	90 ^2^
			IgM	90 ^2^	90 ^2^
COVID 19 IgG/IgM Antibody Rapid Test Kit	Changchun Wancheng Bio-Electron Co., Ltd.	LFA	IgG/IgM	Not specified	Not specified
2019-nCoV IgG/IgM Detection Kit (Colloidal Gold-Based)	Nanjing Vazyme Medical Technology Co., Ltd.	LFA	IgG	96.7 ^2^	90 ^2^
			IgM	66.7 ^2^	77.5 ^2^
SARS-CoV-2 Antibody Test (lateral Flow Method) Wondfo	Guangzhou Wondfo Biotech Co., Ltd.	LFA	Total Antibodies	86.4 [52]	99.6 [52]

CLIA: chemiluminescent immunoassay; LFA: lateral flow assay; ELISA: enzyme-linked immunosorbent assay; ELFA: enzyme-linked fluorescence assay. ^1^ Reported by the company; ^2^ Reported by the Food and Drug Administration (FDA).

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
