# Peer review of "SARS-CoV-2 in Mexico: Beyond Detection Methods, Scope and Limitations"

_diagnostics, 2021, doi:10.3390/diagnostics11010124_

Round 1

Reviewer 1 Report

Rapid and accurate diagnosis of COVID-19 is essential for the clinical managements of SARS-CoV-2 infection as well as pandemic control.  In this manuscript, the authors attempted to describe different SARS-CoV-2 detection methods and their strength and weaknesses. I believe that this manuscript may be benefit from a major revision. 

  1. The authors categorized diagnostic tests based on the direct or indirect detection of the virus and included antigen testing into indirect group, however, antigen tests are indeed direct tests. The discussion of antigen tests should be moved from its current section.
  2. Studies have shown that pre-analytical factors such as specimen types, specimen quality or the collection time are crucial for the performance of a diagnostic assay, especially for molecular diagnosis of COVID-19. It would add value of this manuscript if the authors include the discussion.
  3. RNA viruses evolve over time and changes can happen at the level of nucleotides. Among 7 protocols that were recommended by the WHO early on, some have been shown to have reduced inclusivity due to mismatches of nucleotides between the primers/probes and viral genome (Kevin S. et al. JCV 2020). The statement in the manuscript does not hold true (line 122-124) and should be re-assessed.
  4. Computed tomography has been used to aid the diagnosis of COVID-19, especially early in the pandemic when laboratory diagnostic devices for COVID-19 were not largely available. It has presented promising performance. However, the clinical specificity and the positive predictive value of this diagnostic method are heavily impacted by the disease prevalence which should be discussed.
  5. Many point-of-care tests have been approved by US FDA for emergency use authorization, including molecular tests and antigen tests. The performance of these tests has been evaluated by independent studies. It would add value to this manuscript if the authors include a review of their performance.

Author Response

Reviewer: 1  

Note: According to the modifications, the reviewers decided to change the name of the paper to "SARS-CoV-2 in Mexico: Beyond of detection methods, scope and limitations"

Please see the attached file where the observations are marked.

COMMENTS TO AUTHORS

Point 1: The authors categorized diagnostic tests based on the direct or indirect detection of the virus and included antigen testing into indirect group, however, antigen tests are indeed direct tests. The discussion of antigen tests should be moved from its current section.

Response 1: As the reviewer suggest, antigen detection methods were reorganized into direct method sections according (please see page 3, lines 95,96, and figure 1, panel A).

Point 2: Studies have shown that pre-analytical factors such as specimen types, specimen quality or the collection time are crucial for the performance of a diagnostic assay, especially for molecular diagnosis of COVID-19. It would add value of this manuscript if the authors include the discussion.

Response 2: As the reviewer suggested we expanded the information and discussion about this point in section 7 (please see page 7, lines 232,245), as follow: For accurate viral genetic detection, factors such as the time the biological sample is taken, at a very early or late stage (where the viral load is low), or the quality (degradation of genetic material) and amount of the sample, influence the accurate, sensibility and reproducibility of the result.

Point 3: RNA viruses evolve over time and changes can happen at the level of nucleotides. Among 7 protocols that were recommended by the WHO early on, some have been shown to have reduced inclusivity due to mismatches of nucleotides between the primers/probes and viral genome (Kevin S. et al. JCV 2020). The statement in the manuscript does not hold true (line 122-124) and should be re-assessed.

Response 3: As the reviewer suggested we re-assessed this statement as follow (please see page 5, line 148,149). Although the protocols suggested by the WHO are mentioned, the sensitivity and specificity of each one is not discussed, nor is the use of any one in particular suggested. In this article we only point out that the existing protocols and that the use of each one depends on the institution and the country resource´s availability to perform each molecular test.

Point 4: Computed tomography has been used to aid the diagnosis of COVID-19, especially early in the pandemic when laboratory diagnostic devices for COVID-19 were not largely available. It has presented promising performance. However, the clinical specificity and the positive predictive value of this diagnostic method are heavily impacted by the disease prevalence which should be discussed.

Response 4: This section refers to discussing an alternative for the detection and follow-up of patients in countries with precarious health systems with little capacity to perform complete clinical and laboratory studies for a high number of patients, however, since it is not considered a diagnostic method, the reviewers decided to remove this information from page 7, line 230.

Point 5: Many point-of-care tests have been approved by US FDA for emergency use authorization, including molecular tests and antigen tests. The performance of these tests has been evaluated by independent studies. It would add value to this manuscript if the authors include a review of their performance.

Response 5: The suggested information was included in section 7 (please see page 11 lines 328,337).

Reviewer 2 Report

Martinez-Liu et al in the review ‘SARS-CoV-2: Beyond of Detection Tools and Diagnostics in Mexico’ summarizes the current diagnostics methods for SARS-CoV-2 and their advantages and disadvantages. The review is useful given a large number of methods currently use. However, several points should be addressed before its publication.  The title indicates Mexico as the place where the diagnostics are performed but the connection to Mexico is not obvious in large part of the text and only briefly stated at the end. Which detection technology is mostly used in Mexico, are there different diagnostics strategies? Comparison to diagnostics workflows used for the Swine flu pandemic in Mexico? The focus and specific goals of the review should be outlined already in the abstract or the introduction.

2) Line 17: R0: 2-3 should be explained and perhaps better positioned in the introduction

3) Line 32: remove ‘novel’ it is not necessary, the name is SARS-CoV-2

4) Line 42: The sentence’ of approximately between 60-40 nm size’ does not seem to make sense. SARS-CoV-2 has an average diameter of 90 nm (reference: https://doi.org/10.1038/s41467-020-19619-7)

5) Line 45: Typo: polyprotein 1a/1ab1 should be pp1a and pp1ab

6) Line 47: receptor biding domain (small caps)

7) COVID-19 can manifest in different ways but the multisystemic disease would mean that it always causes many different symptoms. I would suggest removing the sentence:‘Because of that it could be classified as a multisystemic disease.’ unless there is supporting evidence.

8) Line 81: ‘The direct detection test identifies the virus by its genome using molecular methods such as PCR-based assays.’ Immunoassay detecting a viral antigen (e.g. S spike, N protein) from a swab sample is also a direct detection test. In addition RT-LAMP technology is missing (references:  https://doi.org/10.3390/v12080863; DOI: 10.1126/scitranslmed.abc7075)

9) Line 93: sensitivity and specificity terms should be introduced here

10) Figure 1. There is no hemaglutinin acetylesterase present in SARS-CoV-2 virions, please remove it from the figure. RT-LAMP technology is missing.

11) Line 116: It is not clear what ‘OR’ means

12) Line 122: There is a mistake in the sentence: ‘buy’ should be removed.

13) Line 127-129: Please explain what is meant by this sentence. Several publications are showing that saliva can be used for SARS-CoV-2 detection by RT-PCR. Do you mean that this was not yet established in Mexico? It is not clear if section 4 is referring to the situation in Mexico.

14) Line 135: ‘extrapolated’ is not fitting well the sentence. Please modify the sentence to clarify it.

15) Line 152: What is meant here by the “classification of patients” with the disease?

16) Section 5. There are two detection strategies but only host antibody detection is discussed. Please discuss also tests using viral antigens often used in rapid tests.

17) Section 6. Computed tomography of the chest is not a diagnostic method. A diagnostic method must be able to determine a pathogen (specificity), CT only detects lung damage. Moreover, during CT, the individual is irradiated by X-ray and that is invasive, and this technique would likely fail to detect asymptomatic cases. CT of the lung is a certainly important tool to learn about the lung damage in COVID-19 patients but should not be considered as a diagnostic tool since it cannot distinguish between different pathogens or lung diseases. This should be removed.

18) Section 7: Perhaps a better title would be “Interpretation of results obtained by COVID-19 detection methods and their pitfalls”.

19) Line 214: What is meant here by confirmatory RT-PCR, do you mean the second RT-PCR test using another set of primers?

20) Figure 2. Immunoassay detecting viral antigen (e.g. N protein) is missing in figure 2.; Typo: in “Window”.

21) Section 8. Comment: POC is certainly very important but a rapid test must be offering sufficient sensitivity and specificity. POC using antibodies developed by the host or POC using tests detecting viral components (RT-LAMP) or viral antigen immunoassays. Please provide a specific suggestion of how such a diagnostic workflow should look like considering the situation and resources in Mexico. Please briefly also discuss drawbacks or potential problems of POCs.

22) Line 259: typo: (August 28t, 2020).

23) Title section 2: Symptoms and transmission routes of COVID-19 (instead of contagion routes).

24) Line 46: 16 nonstructural proteins instead of 15 (https://www.nature.com/articles/s41579-020-00468-6).

Author Response

We performed all the suggested modifications to the edited manuscript send by the Editor, and in addition we also performed all modification suggested by the reviewers as follow:

Please see the attached file where the observations are marked

Note: According to the modifications, the reviewers decided to change the name of the paper to "SARS-CoV-2 in Mexico: Beyond of detection methods, scope and limitations"

Reviewer: 2  

Point 1: The review is useful given a large number of methods currently use. However, several points should be addressed before its publication. 

Point 1.1: The title indicates Mexico as the place where the diagnostics are performed but the connection to Mexico is not obvious in large part of the text and only briefly stated at the end. Which detection technology is mostly used in Mexico, are there different diagnostics strategies?

Response 1.1: In Mexico confirmed diagnosis of SARS-CoV-2 infection is performed by RT-PCR assay in nasopharingeal sample obtained between the first 5 days upon symptoms appear. Until now there are a clear unified strategy as we describe in figure 3 (please see page 11 lines 300,309)

Point 1.2: Comparison to diagnostics workflows used for the Swine flu pandemic in Mexico?

Response 1.2: In fact, diagnosis strategies to detect Influenza virus are different than the one designed for SARS-CoV-2. Both confirmatory tests, are based on RT-PCR assay, but using different number of genes because the variability and differences of each virus genome are considered.

Point 1.3: The focus and specific goals of the review should be outlined already in the abstract or the introduction.

Response 1.3: As you suggested we added specific goals into section 1 (please see page 2 lines 57,68)

Point 2: Line 17: R0: 2-3 should be explained and perhaps better positioned in the introduction

Response 2: As you suggested, the Ro definition was better explained and relocalized into section 2 (please see page 2, line 81-83).

Point 3: Line 32: remove ‘novel’ it is not necessary, the name is SARS-CoV-2

Response 3: As the reviewer suggested this word was removed from the paragraph (please see page 1 line 33)

Point 4: Line 42: The sentence’ of approximately between 60-40 nm size’ does not seem to make sense. SARS-CoV-2 has an average diameter of 90 nm (reference: https://doi.org/10.1038/s41467-020-19619-7).

Response 4: We performed the suggested correction in the virus size (please see page 1, line 41).

Point 5: Line 45: Typo: polyprotein 1a/1ab1 should be pp1a and pp1ab

Response 5: The correction was performed (please see page 2 line 45).

Point 6: Line 47: receptor biding domain (small caps)

Response 6: We performed suggested correction (please see page 2, line 47,48).

Point 7: COVID-19 can manifest in different ways but the multisystemic disease would mean that it always causes many different symptoms. I would suggest removing the sentence: ‘Because of that it could be classified as a multisystemic disease.’ unless there is supporting evidence.

Response 7: We wanted to indicate that clinical manifestations involve many different organs in the body. But as you suggested we removed the sentence (page 2, line 56).

Point 8: Line 81: ‘The direct detection test identifies the virus by its genome using molecular methods such as PCR-based assays.’ Immunoassay detecting a viral antigen (e.g. S spike, N protein) from a swab sample is also a direct detection test. In addition RT-LAMP technology is missing  (references:  https://doi.org/10.3390/v12080863; DOI: 10.1126/scitranslmed.abc7075)

Response 8: Antigen detection was relocated to direct methods section (please see page 3, line 95-96) and we also include information about the use of these tests in Mexico (please see page 8, lines 288,290) and their major characteristics (please see page 11, lines 328,337). The scope of this review involves only those detection tests that are available in Mexico, because of that, RT-LAMP was not included in detail. But we added minimal information about this technology and the reference (please see page 12, line 339,353).

Point 9: Line 93: sensitivity and specificity terms should be introduced here

Response 9: As the reviewer suggest, we added information about these two terms (please see page 3, lines 108-115).

Point 10: Figure 1. There is no hemaglutinin acetylesterase present in SARS-CoV-2 virions, please remove it from the figure. RT-LAMP technology is missing.

Response 10: You are correct, and we removed this enzyme from the figure (please see edited figure 1, line 122).

Point 11: Line 116: It is not clear what ‘OR’ means

Response 11: Sorry, you are right, we completed the meaning into the manuscript (please see page 2, line 45) and correct the word (please see page 5 line 142).

Point 12: Line 122: There is a mistake in the sentence: ‘buy’ should be removed.

Response 12: We performed the suggested correction (please see page 5, line 148).

Point 13: Line 127-129: Please explain what is meant by this sentence. Several publications are showing that saliva can be used for SARS-CoV-2 detection by RT-PCR. Do you mean that this was not yet established in Mexico? It is not clear if section 4 is referring to the situation in Mexico.

Response 13: As the reviewer suggest we added more information to clarify the status of this procedure using saliva in Mexico. (Please see page 5, lines 151,155)

Point 14: Line 135: ‘extrapolated’ is not fitting well the sentence. Please modify the sentence to clarify it.

Response 14: As you suggested we rewrite this sentence (plese see page 6, lines 164,166).

Point 15: Line 152: What is meant here by the “classification of patients” with the disease?

Response 15: We just tried to speculate about the possible relationship between the title of Anti-SARS-CoV-2 IgM  with clinical manifestation; but you are rigth, until now this assumption can not be established. So, we prefer to remove this confusing statement. (See page 6, lines 181).

Point 16: Section 5. There are two detection strategies but only host antibody detection is discussed. Please discuss also tests using viral antigens often used in rapid tests.

Response 16: As the reviewer suggest we include information of antigens detection method and their major characteristics (please see page 11, lines 328,337).

Point 17: Section 6. Computed tomography of the chest is not a diagnostic method. A diagnostic method must be able to determine a pathogen (specificity), CT only detects lung damage. Moreover, during CT, the individual is irradiated by X-ray and that is invasive, and this technique would likely fail to detect asymptomatic cases. CT of the lung is a certainly important tool to learn about the lung damage in COVID-19 patients but should not be considered as a diagnostic tool since it cannot distinguish between different pathogens or lung diseases. This should be removed.

Response 17: You are right, so we are in agreement, then we already remove this information from page 7, line 230.

Point 18: Section 7: Perhaps a better title would be “Interpretation of results obtained by COVID-19 detection methods and their pitfalls”.

Response 18: We consider the new title you proposed, adding the context in which we will approach it (only in Mexico) (please see page 7 line 230).

Point 19: Line 214: What is meant here by confirmatory RT-PCR, do you mean the second RT-PCR test using another set of primers?

Response 19: No, there is not a second one confirmatory test. We wanted to remark that the only confirmatory test accepted worldwide is RT-PCR.

Point 20: Figure 2. Immunoassay detecting viral antigen (e.g. N protein) is missing in figure 2.; Typo: in “Window”.

Response 20: As the reviewer suggest, we added EIA viral antigen detection in figure 2.

Point 21: Section 8. Comment: POC is certainly very important but a rapid test must be offering sufficient sensitivity and specificity. POC using antibodies developed by the host or POC using tests detecting viral components (RT-LAMP) or viral antigen immunoassays. Please provide a specific suggestion of how such a diagnostic workflow should look like considering the situation and resources in Mexico. Please briefly also discuss drawbacks or potential problems of POCs.

Response 21: We review the available literature about this point, and we added specific information about  POC  availability, use and meaning into the changing diagnostic environment in Mexico. (please see page 11 lines 339,353).

Point 22: Line 259: typo: (August 28t, 2020).

Response 22: The suggested correction was performed (page 8 line 277).

Point 23: Title section 2: Symptoms and transmission routes of COVID-19 (instead of contagion routes).

Response 23: The suggested correction was performed (page 2 line 69)

Point 24: Line 46: 16 nonstructural proteins instead of 15 (https://www.nature.com/articles/s41579-020-00468-6).

Response 24: The suggested correction was performed (page 2 line 45).

Round 2

Reviewer 1 Report

The authors have addressed my comments. I recommend to accept the manuscript.